# Mice Microbiota Composition Changes by Inulin Feeding with a Long Fasting Period under a Two-Meals-Per-Day Schedule

**DOI:** 10.3390/nu11112802

**Published:** 2019-11-16

**Authors:** Hiroyuki Sasaki, Hiroki Miyakawa, Aya Watanabe, Yuki Nakayama, Yijin Lyu, Koki Hama, Shigenobu Shibata

**Affiliations:** 1Laboratory of Physiology and Pharmacology, School of Advanced Science and Engineering, Waseda University, Shinjuku-ku, Tokyo 162-8480, Japan; hiroyuki-sasaki@asagi.waseda.jp (H.S.); hgbbst-hiroki@toki.waseda.jp (H.M.); aya_watanabe7115@suou.waseda.jp (A.W.); yukibecky-6991@akane.waseda.jp (Y.N.); ikin@fuji.waseda.jp (Y.L.); koby_frekoi418@ruri.waseda.jp (K.H.); 2AIST-Waseda University Computational Bio Big-Data Open Innovation Laboratory (CBBD-OIL), National Institute of Advanced Industrial Science and Technology, Shinjuku-ku, Tokyo 169-8555, Japan

**Keywords:** microbiota, inulin, circadian rhythm, feeding timing

## Abstract

Water-soluble dietary fiber is known to modulate fecal microbiota. Although there are a few reports investigating the effects of fiber intake timing on metabolism, there are none on the effect of intake timing on microbiota. Therefore, in this study, we examined the timing effects of inulin-containing food on fecal microbiota. Mice were housed under conditions with a two-meals-per-day schedule, with a long fasting period in the morning and a short fasting period in the evening. Then, 10–14 days after inulin intake, cecal content and feces were collected, and cecal pH and short-chain fatty acids (SCFAs) were measured. The microbiome was determined using 16S rDNA sequencing. Inulin feeding in the morning rather than the evening decreased the cecal pH, increased SCFAs, and changed the microbiome composition. These data suggest that inulin is more easily digested by fecal microbiota during the active period than the inactive period. Furthermore, to confirm the effect of fasting length, mice were housed under a one-meal-per-day schedule. When the duration of fasting was equal, the difference between morning and evening nearly disappeared. Thus, our study demonstrates that consuming inulin at breakfast, which is generally after a longer fasting period, has a greater effect on the microbiota.

## 1. Introduction

In the gut of mammals, the microbiota includes 100 trillion bacteria. Disordered microbiota alteration is involved in the development of various diseases [1]. *Firmicutes* are bacteria related to obesity, while *Bacteroidetes* suppress fat accumulation in mice fed a high-fat diet (HFD) [2]. When the feces of obese mice are transplanted into germ-free mice, obesity develops [3]. Moreover, *Fusobacterium*, including *Fusobacterium nucleatum*, are increased in patients with colorectal cancer compared with healthy subjects [4,5]. In addition to physical illnesses, a relationship of microbiota with psychological illness has also been reported. In patients with major depression, *Bacteroidetes*, *Proteobacteria*, and *Actinobacteria* are significantly increased compared with healthy subjects [6]. These results suggest that intestinal bacteria are related to the development of diseases and that maintaining homeostasis of the microbiota is important for the mental and physical health of the host.

Short-chain fatty acids (SCFAs) are produced when the microbiota ferments and degrades non-digestible food components [7]. The SCFAs lower intestinal pH, suppress the growth of pathogenic bacteria in the gut, and function as a regulator of metabolism and immunity [8]. Among SCFAs, acetic acid is a liver energy substrate used for fat synthesis, and propionic acid is used as a material for gluconeogenesis in the liver. Butyrate promotes the induction of regulatory T cells in the large intestine [9,10]. Furthermore, SCFAs also increase insulin sensitivity in the liver and muscles through GPR43, a receptor for SCFAs in white fat, as well as increase energy efficiency [11].

The circadian rhythm, controlled by clock genes, plays an important role in daily locomotor activity rhythms and physiological events, such as the sleep–wake cycle, hormone secretion, and the sympathetic nervous system [12,13]. Clocks in peripheral tissues are regulated by the central clock in the suprachiasmatic nucleus and external cues such as food, temperature, and exercise [14,15,16]. It has been reported that circadian rhythms are also present in the intestinal flora and controlled by dietary composition [17,18,19,20]. Furthermore, disturbance of the circadian clock due to jet lag alters microbial populations. For example, when the stool of jet-lagged mice is transplanted into germ-free mice, the recipient mice become obese [19]. Recently, however, it has been reported that SCFAs produced by gut microbiota can synchronize the circadian clock [21].

The microbiota composition changes depending on food components. In particular, foods rich in dietary fiber have a strong effect on the microbiota and are known as prebiotics [22]. According to Gibson et al., prebiotics are defined as “nondigestible food ingredients that beneficially affect the host by selectively stimulating the growth and/or activity of one or a limited number of bacterial species already resident in the colon, and thus attempt to improve host health” [23]. Inulin is a water-soluble dietary fiber and, thus, a prebiotic. It is particularly involved in the growth of bacteria that produce lactic acid [24] and promote the absorption of minerals such as calcium and magnesium [25,26].

It has been suggested that meal timing and daily eating habits may affect the development and prevention of lifestyle-related diseases such as obesity. A study by Hatori et al. demonstrates that restricted feeding in an activity period for mice without reducing calorific intake prevents metabolic diseases in mice fed a HFD [27]. Mice consuming milk fat late in the activity period have elevated hepatic fat and increased serum triglycerides and free fatty acids [28]. In addition, scheduled access to a HFD during the inactivity period increases body weight in mice compared with access during the activity period [29,30]. Moreover, in human experiments, the combination of a late dinner with a short sleep duration is associated with the risk of obesity [31]. In addition, the risk of obesity has been related to eating supper after 20:00 in the evening [32]. In recent years, it has been suggested that the influence of food on lipid metabolism is different depending on the time of food intake. In mice fed a high-fructose diet, fish oil given earlier in the activity period rather than later more effectively lowered lipids [33].

There are many reports indicating that time of food intake affects energy metabolism, but there are still relatively few reports describing the effect of eating time on microbiota. Furthermore, there are few reports on the dual effect of food type, particularly dietary fiber, and intake time on microbiota. Therefore, in the current study, we investigated whether inulin intake during the morning has a stronger effect on the microbiota than inulin intake during the evening with a two-meals-per-day schedule in mice.

## 2. Materials and Methods

### 2.1. Mice

In this study, we used eight-week-old male ICR mice (Tokyo Laboratory Animals, Tokyo, Japan). The mice were kept in a room maintained on a 12 h light/12 h dark (LD) cycle (lights on from 08:00 to 20:00). Zeitgeber time 0 (ZT0) was designated as lights-on time and ZT12 as lights-off time under the LD cycle. The mice were housed either in groups (five mice per cage; experiments 1 and 2) or individually (experiments 3 and 4) in plastic cages. The cages were maintained at a temperature of 22 ± 2 °C, humidity of 60 ± 5%, and light intensity of 100–150 lux. The mice were provided with a HFD containing 45% kcal of fat (Diet 12451; Research Diets Inc., New Brunswick, NJ, USA) with cellulose (Oriental Yeast Co., Ltd., Tokyo, Japan) or inulin (Fuji FF; Fuji Nihon Seito Co., Tokyo, Japan) [34,35] and water ad libitum. This HFD is a diet used as a model for obesity, diabetes, and fatty liver in rodents [36,37]. Inulin has been reported to attenuate HFD-induced lipid metabolism and microbiota change [38]. In addition, the metabolic syndrome caused by obesity and abnormal lipid metabolism in the liver are related to microbiota change [39,40,41]. Therefore, we used an HFD to enhance the attenuating effects of inulin with the condition of microbiota change. The animal experiment was conducted with permission from the Committee for Animal Experimentation of the School of Science and Engineering at Waseda University (permission # 09A11, 10A11) and in accordance with the law (No. 105) and notification (No. 6) of the Japanese government.

### 2.2. Scheduled Feeding

We prepared two types of feeding conditions. In type 1 (experiments 1 and 2), only the feeding time was controlled, while in type 2 (experiments 3 and 4), both the start time of feeding and the amount of food were controlled.

In type 1 feeding, all of the mice could approach the feed box during the permitted time. We defined the morning as ZT12–20 and the evening as ZT20–4. The mice had free access to the feed box for predetermined four-hour periods (morning meal as ZT12–16 and evening meal as ZT20–0). Throughout the remaining time, the feed box was locked. Food intake was calculated by measuring the weight of the food in the feed box at the start and end of the experiment. The total consumed food was divided by the number of mice and the number of days in the experiment. In the type 1 experiments, we housed the mice as a group to avoid the stress induced by individual housing.

In type 2 feeding, all of the mice were housed in cages containing food dispensers that released food pellets under the regulation of a timer. The mice were fed 90% of the amount of food that was consumed in experiment 1 (Figure 1c). In experiment 3, the mice were fed two meals per day at ZT12 (defined as morning) or ZT20 (defined as evening); the meal size was 1.8 g. In experiment 4, the mice were fed one meal per day at ZT12 (morning) or ZT20 (evening); the meal size was 3.6 g.

We adjusted the concentration of dietary fiber so that the amount of inulin was approximately equal between experiments.

### 2.3. Cecal pH Measurement

The cecal pH was measured by inserting the glass tip of an electrode of a pH meter (pH Spear; Eutech Instruments, Vernon Hills, IL, USA) directly into the cecum.

### 2.4. Measurement of SCFAs

The SCFAs were measured via gas chromatography and flame ionization detection (Shimadzu Corp., Kyoto, Japan) as described by a previous report [42] with some modifications. A total of 0.05 g of cecal content was acidified with 0.05 mL sulfuric acid (FUJIFILM Wako Pure Chemical Corp., Osaka, Japan). Then, the SCFAs were extracted by shaking with 0.4 mL of diethyl ether (FUJIFILM Wako Pure Chemical Corp., Osaka, Japan) and 0.2 mL of ethanol (FUJIFILM Wako Pure Chemical Corp., Osaka, Japan), which was then centrifuged at 14,000 rpm at room temperature for 30 s. A total of 1 μL of the organic phase was injected into the capillary column (InertCap Pure WAX (30 m × 0.25 mm, df = 0.5 μm); GL Science Inc., Tokyo, Japan). The initial temperature was 80 °C, and the final temperature was 200 °C. Helium was used as a carrier gas. Quantification of the samples was performed using calibration curves for acetic, lactic, propionic, and butyric acids. A standard curve for each acid was conducted for their quantitation in the samples.

### 2.5. Fecal DNA Extraction

The fecal DNA was extracted as previously described, with some modifications [43]. About 0.2 g of the fecal sample was suspended in a 50 mL Falcon tube containing 20 mL PBS. The suspension was filtered through a 100-μm mesh nylon filter (Corning Inc., New York NY, USA). The debris on the filter was washed with 10 mL of Phosphate buffered salts (PBS). The filtrates were centrifuged at 4000 rpm for 20 min at 4 °C, and each precipitate was then suspended with 1.5 mL of TE10 buffer (10 mM Tris-HCl (FUJIFILM Wako Pure Chemical Corp., Osaka, Japan)/10 mM ethylenediaminetetraacetic acid (EDTA) (DOJINDO, Tokyo, Japan)). The suspensions were transferred to 2-mL microtubes before being centrifuged at 10,000 rpm for five minutes at 4 °C. Following this, each precipitate was suspended again with 0.8 mL of TE10 buffer. The DNA was extracted using 1 mL of PCI (Invitrogen, Carlsbad, CA, USA) and isolated with 0.1 mL of lysozyme (FUJIFILM Wako Pure Chemical Corp., Osaka, Japan) and 0.02 mL of achromopeptidase (FUJIFILM Wako Pure Chemical Corp., Osaka, Japan). The DNA was purified via treatment with RNase (Promega Corp., Madison, WI, USA), followed by precipitation with 20% PEG solution (Tokyo Chemical Industry Co., Ltd., Tokyo, Japan). Finally, the DNA was rinsed with 70% ethanol and dissolved in 50 μL TE buffer.

### 2.6. 16 S rDNA Gene Sequencing

The 16S rDNA gene sequencing was performed according to the instructions of Illumina. V3–V4 variable regions of the 16S rDNA gene were amplified by PCR using the following primers: forward primer = 5′-TCGTCGGCAGCGTCAGATGTGTATAAGAGACAGCCTACGGGNGGCWGCAG-3′;reverse primer = 5′-GTCTCGTGGGCTCGGAGATGTGTATAAGAGACAGGACTACHVGGGTATCTAATCC-3′.

The PCR amplification was performed with 2.5 μL microbial DNA (5 ng/μL), 5 μL of each primer (1 μmol/L), and 12.5 μL 2 × KAPA HiFi HotStart Ready Mix (Kapa Biosystems Inc., Wilmington, MA, USA). The following PCR procedure was used: 95 °C for three minutes, followed by 25 cycles of 95 °C for 30 s, 55 °C for 30 s, and 72 °C for 30 s. Finally, an extension was performed at 72 °C for five minutes. The Amplicon PCR products were purified using AMPure XP beads (Beckman Coulter, Inc., Brea, CA, USA), according to the manufacturer’s instructions. A Nextera XT Index Kit v2 (Illumina Inc., Hayward, CA, USA) was used for the Illumina sequencing adapters and attachment of the dual indices. An index PCR was performed with 5.0 μL PCR product, 5.0 μL of each of the Nextera XT Index Primers, 25 μL 2× KAPA HiFi HotStart Ready Mix, and 10 μL PCR-Grade Water. The PCR was performed via the following procedure: 95 °C for three minutes, followed by eight cycles of 95 °C for 30 s, 55 °C for 30 s, and 72 °C for 30 s. Finally, an extension was performed at 72 °C for five minutes. The index PCR products were purified using AMPure XP beads (Beckman Coulter, Inc., Brea, CA, USA). The quality of the purifications was checked using the Agilent 2100 Bioanalyzer with a DNA1000 Kit (Agilent Technologies Inc., Santa Clara, CA, USA). Finally, the DNA library was diluted to 4 nmol/L.

Then, the DNA library was sequenced using the Miseq Reagent Kit v3 (Illumina Inc.) in the Illumina Miseq 2 × 300 bp platform, according to the manufacturer’s instructions.

### 2.7. Analysis of 16S rDNA Gene Sequences

The 16S rDNA sequence reads were processed by the Quantitative Insights into Microbial Ecology (QIIME) pipeline version 1.9.1 [44]. The quality-filtered sequence reads were assigned to operational taxonomic units at 97% identity with the UCLUST algorithm [45]; these reads were then compared with reference sequence collections in the Greengenes database (August 2013 version). A total of 6,680,549 reads were obtained from the 91 samples. On average, 73,412 ± 4606 reads were obtained per sample. The taxonomy summary at the phylum to genus levels, alpha diversity such as the Simpson diversity index, beta diversity, and principal coordinate analysis (PCoA) were calculated and generated using QIIME. A PCoA analysis was also calculated using weighted UniFrac distances.

### 2.8. Predicted Metagenomes

In experiments 3 and 4, the functional profiles of microbial communities were predicted by the Phylogenetic Investigation of Communities by Reconstruction of Unobserved States (PICRUSt) [46]. The functional predictions were assigned to the Kyoto Encyclopedia of Genes and Genomes (KEGG) ortholog functional profiles of microbial communities via 16S sequences. We selected and examined categories related to “carbohydrate metabolism” for simplification and clarity of the analysis.

### 2.9. Statistical Analysis

The data were expressed as means ± standard error of the mean (SEM). All statistical analyses were performed using GraphPad Prism (version 6.03, GraphPad Software Inc., San Diego, CA, USA). We checked whether the data showed a normal or non-normal distribution and equal or biased variation via the D’Agostino-Pearson test/Kolmogorov–Smirnov test and F-value test/Bartlett’s test, respectively. If the data showed a normal distribution and equal variation, the statistical significance was determined by the Student’s *t*-test or one-way ANOVA with a Tukey’s test or two-way ANOVA with a Tukey’s post-hoc analysis if the interaction was significant. If the interaction was not significant but the main effect was, Sidak’s post-hoc analysis was used. If the data showed a non-normal distribution or biased variation, the statistical significance was determined by the Mann–Whitney test or Kruskal–Wallis test with a Dunn post-hoc analysis and a two-stage linear step-up procedure of the Benjamini, Krieger, and Yekutieli test for multiple comparisons. The permutational multivariate analysis of variance (PERMANOVA) was used to assess the change of the microbiota composition. The PERMANOVA was analyzed by QIIME.

## 3. Results

### 3.1. Inulin Intake Changed Microbiota Composition under Both Morning and Evening Timings

In this experiment, cellulose, an insoluble dietary fiber, was added to the HFD as a control for inulin because the dietary fiber contained in this HFD was cellulose. The amount of dietary fiber in the food was kept the same between each group. Indeed, when comparing the HFD with and without cellulose, there was no significant difference in the body weight, food intake, cecal pH, amount of SCFAs, or microbiota composition of the mice (Appendix A). Therefore, this concentration of cellulose did not appear to affect these factors.

Some reports have suggested that inulin consumption induces changes in the microbiota composition [47,48,49,50]. Here, we divided the mice into two groups. Group 1 received cellulose and was fed an HFD with 2.5% cellulose in the morning and evening. Group 2 received inulin and was fed HFD with 2.5% inulin in the morning and evening. The mice were housed under each condition for 10 days, after which they were sacrificed at ZT20 (four hours after the morning intake) or ZT4 (four hours after the evening intake) on days 10–11 (Figure 1a). We sampled the cecal content and feces from the rectum and measured the cecal pH. There was no significant difference in the body weight between the two groups before sacrifice (Figure 1b), nor was there was a difference in the food intake between them (Figure 1c). There were no standard error bars in the food intake volume because of the group housing.

The cecal pH was significantly lower in the inulin group than in the cellulose group at both ZT20 and ZT4 (Figure 1d). The propionic acid level was significantly higher in the inulin group than in the cellulose group at ZT4, and the lactic acid level was significantly higher in the inulin group than in the cellulose group at ZT20. At ZT4, there was only a slight increase in the inulin group compared with the cellulose group. In the cellulose group, the lactic acid level was significantly different between ZT20 and ZT4. There were no significant differences in the acetic acid, propionic acid, and total SCFA levels between the cellulose and inulin group at either ZT20 or ZT4 (Figure 1e–i).

As the propionic and lactic acid increased and the cecal pH decreased, the microbiota may have changed due to inulin feeding. Therefore, we extracted 16S rDNA from the mice feces and analyzed the microbiota. In the cellulose group, the values of alpha-diversity as described by the Simpson index were significantly higher at ZT4 than at ZT20. The Simpson index was significantly higher in the inulin group than in the cellulose group at ZT20, but there was no significant difference observed at ZT4 (Figure 1j). Next, we examined the differences in the changes of the relative abundance of taxa between the inulin group and cellulose group. Some of the detected bacteria are shown in Figure 2. At the phylum level, the relative abundance of *Firmicutes* was significantly lower in the inulin group than in the cellulose group at ZT20. However, there was no significant difference in the relative abundance of *Bacteroidetes* between the inulin and cellulose groups, though the levels were increased slightly in the former (Figure 2a). At the genus level, the relative abundance of *Lactococcus* and *Streptococcus* significantly decreased in the inulin group at ZT20, and the relative abundance of *Oscillospira* significantly decreased in the inulin group at ZT4 (Figure 2b). We analyzed the PCoA of the weighted UniFrac distances and determined the beta-diversity of the microbiota composition (Appendix A). In this experiment, we focused on the influence of inulin on the microbiota; thus, we primarily compared cellulose and inulin feeding. The beta-diversity of the microbiota composition was significantly different between the cellulose and inulin groups at ZT20 but not at ZT4 (Appendix A).

These results suggest that inulin consumption changes microbiota composition. In addition, the inulin feeding time may have different effects on the microbiota because changes in the microbiota were more prominent at ZT20 (morning) than at ZT4 (evening), which showed significant and non-significant differences, respectively, compared with the cellulose group.

### 3.2. Inulin Intake in the Morning Rather than the Evening Strongly Affected the Microbiota Composition under Time-Restricted Feeding Conditions

In this study, inulin may have had different effects on the microbiota depending on the feeding times. However, in experiment 1, we did not measure the effect of the feeding pattern. It is possible that the effect of inulin was increased at ZT20 (four hours after the morning intake) due to the high consumption in the morning. In the next experiment, we examined whether morning or evening inulin feeding affected the microbiota under the two meals-per-day schedule. The mice were divided into three groups. Group 1 received cellulose and was fed an HFD with 5% cellulose in the morning and evening. Group 2 received inulin in the morning and was fed an HFD with 5% inulin in the morning and an HFD with 5% cellulose in the evening. Group 3 received inulin in the evening and was fed an HFD with 5% cellulose in the morning and an HFD with 5% inulin in the evening. The mice were housed under each condition for 10 days, after which they were sacrificed at ZT20 and ZT4 on days 10–11 (Figure 3a). We sampled cecal contents and feces and measured the cecal pH. There was no significant difference in body weight between any group before sacrifice (Figure 3b), nor was there a large difference in total food intake between them. However, the total food intake was slightly higher if inulin intake was in the morning rather than in the evening (Figure 3c). The cecal pH was significantly lower in the morning inulin group than in the morning cellulose or evening inulin groups at ZT20. On the contrary, the pH was significantly lower in the evening inulin group than in the evening cellulose and morning inulin groups at ZT4 (Figure 3d). The acetic acid, propionic acid, lactic acid, butyric acid, and total SCFA levels were significantly higher in the morning inulin group than in the morning cellulose or evening inulin groups at ZT20. However, the acetic acid, propionic acid, lactic acid, butyric acid, and total SCFA levels were significantly higher in the evening inulin group than in the evening cellulose or morning inulin groups at ZT4 (Figure 3e–i).

Next, we extracted 16S rDNA from the mice feces and analyzed the microbiota. The value of alpha-diversity as determined by the Simpson index was significantly higher in the morning inulin group than in the morning cellulose or evening inulin groups at both ZT20 and ZT4 (Figure 3j). We also examined the differences between the changes of the relative abundance of taxa between the inulin and cellulose groups. Some of the detected bacteria are shown in Figure 4. At the phylum level, the relative abundance of *Bacteroidetes* was significantly higher in the morning inulin group than in the morning cellulose group at ZT20. Meanwhile, the relative abundance of *Firmicutes* was significantly lower in the morning inulin group than in the morning cellulose and evening inulin groups at ZT20 as well as significantly lower in the evening inulin group than in the evening cellulose group at ZT4 (Figure 4a). At the genus level, the relative abundance of *Lactococcus* was significantly decreased in the morning inulin group at both ZT20 and ZT4 and in the evening inulin group at ZT4, while the relative abundance of *Dorea* and *Allobaculum* was significantly increased in the morning inulin group (Figure 4b). We analyzed the PCoA of the weighted UniFrac distances and determined the beta-diversity of the microbiota composition (Appendix A). At ZT20, the beta-diversity of the microbiota was significantly different between the cellulose and morning inulin groups and the morning and evening inulin groups (Appendix A). At ZT4, the beta-diversity of the microbiota composition was significantly different among all of the groups (Appendix A).

These results suggest that morning inulin feeding affected the microbiota more than evening inulin feeding. However, the inulin intake was higher in the morning inulin group. Therefore, the increased consumption of the morning inulin group may have had more of an impact on the microbiota. To eliminate the effects of different food intakes, we prepared an apparatus to supply equal food amounts at two meals per day in the next experiment.

### 3.3. Inulin Feeding in the Morning Affected the Microbiota Composition More than that in the Evening under Restricted Food Amount Conditions

In this experiment, we provided the mice with two meals per day of 1.8 g of food at ZT12 (morning) and ZT20 (evening) to achieve equal food intake. The mice were divided into three groups. Group 1 received cellulose and was fed 1.8 g of an HFD with 5% cellulose in both the morning and evening. Group 2 received inulin in the morning and was fed 1.8 g of an HFD with 5% inulin in the morning and 1.8 g of an HFD with 5% cellulose in the evening. Group 3 received inulin in the evening and was fed 1.8 g of an HFD with 5% cellulose in the morning and 1.8 g of an HFD with 5% inulin in the evening. The mice were housed under each condition for 14 days, after which they were sacrificed at ZT20 and ZT4 on days 14–15 (Figure 5a). We sampled cecal content and feces and measured the cecal pH. There was no significant difference in body weight between any group before sacrifice (Figure 5b). The cecal pH was significantly lower in the morning inulin group than in the morning cellulose group and significantly lower in the evening inulin group than in the evening cellulose group. Moreover, the cecal pH was significantly lower in the morning inulin group than in the evening inulin group (Figure 5c). The propionic acid, lactic acid, butyric acid, and total SCFA levels were significantly higher in the morning inulin group than in the morning cellulose group, while the propionic acid level was significantly higher in the evening inulin group than in the evening cellulose group (Figure 5d–h).

Next, we extracted 16S rDNA from the mice feces and analyzed the microbiota. The value of alpha-diversity as determined by the Simpson index showed no significant difference between any group (Figure 5i). We also examined the differences in the changes of the relative abundance of taxa. Bacteria detected in over half of all samples are shown in Table 1. At the phylum level, the relative abundance of *Proteobacteria* was significantly increased in the morning inulin group, while the relative abundance of *TM7* was significantly decreased in the morning and evening inulin groups (Table 1a). At the genus level, the relative abundance of *Butyricimonas* was significantly increased in the morning inulin group, while the relative abundance of *AF12*, *Staphylococcus*, *Lactococcus*, *Oscillospira*, *Bilophila*, and *Desulfovibrio* was significantly decreased in the morning inulin group. Meanwhile, the relative abundance of *AF12*, *Odoribacter*, and *Oscillospira* was significantly decreased in the evening inulin group (Table 1b). The number of bacteria changed by inulin feeding in the morning was higher than that changed by inulin feeding in the evening. We analyzed the PCoA of the weighted UniFrac distances and determined the beta-diversity of the microbiota composition (Appendix A). The beta-diversity of the microbiota was significantly different between the cellulose group and morning inulin group (Appendix A), but no significant difference was observed between the cellulose group and evening inulin group (Appendix A). We predicted the functional profiles from sequencing data with PICRUSt. Among the categories related to “carbohydrate metabolism”, the relative abundance of fructose and mannose metabolism was significantly increased in the morning inulin group but not in the evening inulin group (Appendix A).

These results suggest that inulin feeding in the morning may affect the microbiota, even if the food intake amount is the same in the morning and evening.

### 3.4. A Relationship Was Observed between the Length of Fasting Time and Inulin Feeding Stimulation

In experiment 3, it was observed that inulin intake in the morning may have an effect on the microbiota and that the fasting time factored into this effect in the morning. The morning inulin group fasted for 16 h after the previous feeding, while the evening inulin group fasted for 8 h after the previous feeding, meaning that the time until breakfast was longer than the time until dinner. Therefore, the difference in the length of fasting time may have changed the effect on the microbiota. To test this hypothesis, we prepared an experiment with equal fasting times based on one meal a day, in which 3.6 g of food was given to the mice at either ZT12 (morning) or ZT20 (evening). The mice were divided into four groups. Group 1 received cellulose in the morning and was fed 3.6 g of an HFD with 5% cellulose in the morning. Group 2 received inulin in the morning and was fed 3.6 g of an HFD with 5% inulin in the morning. Group 3 received cellulose in the evening and was fed 3.6 g of an HFD with 5% cellulose in the evening. Group 4 received inulin in the evening and was fed 3.6 g of an HFD with 5% inulin in the evening. The mice were housed under each condition for 14 days, after which they were sacrificed at ZT20 and ZT4 on days 14–15 (Figure 6a). We sampled cecal content and feces and measured the cecal pH. The body weight was significantly increased in the evening cellulose and inulin groups compared with the morning cellulose and inulin groups. (Figure 6b). The cecal pH was significantly lower in the morning and evening inulin groups than in the morning and evening cellulose groups (Figure 6c). The propionic and lactic acid levels were significantly higher in the morning inulin group than in the morning cellulose group. In addition, the butyric acid level was higher, albeit not significantly, in the morning inulin group than in the morning cellulose group. Meanwhile, the lactic and butyric acid levels were significantly higher in the evening inulin group than in the evening cellulose group, and the propionic acid level was higher, albeit not significantly, in the evening inulin group than in the evening cellulose group (Figure 6d–h).

Next, we extracted 16S rDNA from the mice feces and analyzed the microbiota. The value of alpha-diversity as determined by the Simpson index was significantly higher in the morning cellulose group than in the evening cellulose group (Figure 6i). We also examined the differences in the changes of the relative abundance of taxa. Bacteria detected in over half of all samples are shown in Table 2. At the phylum level, the relative abundance of *Actinobacteria* was increased in the morning inulin group, but there was no significant difference in the relative abundance in the evening inulin group (Table 2a). At the genus level, the relative abundance of *Bifidobacterium* and *Allobaculum* was significantly increased in the morning inulin group, while the relative abundance of *Streptococcus*, *Oscillospira*, and *Ruminococcus* was significantly decreased in the morning inulin group. Meanwhile, the relative abundance of *Dorea* and *Allobaculum* was significantly increased in the evening inulin group, and the relative abundance of *Staphylococcus* and *Lactococcus* was significantly decreased in the evening inulin group (Table 2b). The number of bacteria changed by inulin feeding in either the morning or the evening was similar. We analyzed the PCoA of the weighted UniFrac distances and determined the beta-diversity of the microbiota composition (Appendix A). The beta-diversity of the microbiota was not significantly different between the cellulose and inulin groups in either the morning or evening (Appendix A). We predicted the functional profiles from sequencing data with PICRUSt. Among the categories related to “carbohydrate metabolism”, the relative abundance of fructose and mannose metabolism was not significantly different between the cellulose and inulin groups in either the morning or evening (Appendix A). These results suggest that inulin intake in either the morning or evening with equal fasting periods does not change microbiota beta-diversity.

## 4. Discussion

In this study, inulin intake changed the composition and profile of the gut microbiota, increased SCFAs, and decreased the cecal pH (Figure 1, Figure 2 and Appendix A). SCFAs are important for health because they improve energy metabolism in the liver and muscles and immune function in the large intestine [9,10,11]. In addition, the effect of inulin on the microbiota was dependent on the timing of inulin intake. Therefore, we gave inulin to the mice in either the morning or evening. The microbiota was more affected by inulin feeding in the morning than in the evening (Figure 5 and Appendix A) because the fasting period was longer for the latter. There has been previous research on fasting time and dietary effects. Previous studies examining postprandial glucose metabolism have shown that breakfast, rather than dinner, can suppress postprandial hyperglycemia and that one of the primary factors is the difference in fasting time [51]. Additionally, in a previous study examining the circadian clock, a meal after a long fasting period strongly synchronized the peripheral clock [52,53]. Under a two-meals-per-day schedule in mice, the same amount of chow after 16 h of fasting could reset the *Per2* gene expression rhythm in the liver clock compared with the same amount of chow after 6 h of fasting; in the two-meal experiments presented here, we used exactly the same protocol. In the current experiment, there was no difference in the cecal pH or SCFAs measurements between morning and evening with the same fasting duration (Figure 6c,h). Considering actual human life, the fasting time until breakfast is generally the longest among the three meals. Thus, these results, along with those of the previous study [53], support that inulin intake in the morning is most effective at attenuating HFD-induced changes of the gut microbiota. However, since the gut microbiota is also related to the circadian clock, there may be a difference between morning and evening in the gut microbiota composition, regardless of the fasting time. In addition to the daily feeding model used in this study, a feeding model for equalizing fasting time has been considered [52,54,55]. By using these feeding models, the relationship between fasting time and the effects of foods may be clarified. Moreover, the feeding model of this study has too long a starvation period compared with actual human life. Therefore, a feeding model that mimics the actual human lifestyle of three meals a day, as reported by Kuroda et al., may be considered for future experiments [52].

In this study, we first regulated the access time to inulin-containing food under a two-meals-per-day schedule because mice access food in the morning rather than in the evening under ad-lib food conditions [54,56]. Under these feeding conditions, we found clear effects of inulin in the morning. Therefore, in ad-lib feeding conditions, functional food intake at an earlier time during the active period may be a considerable factor in microbiota changes. Next, we regulated the food volume under a two-meals-per-day schedule. Once again, inulin in the morning had a clear effect on the microbiota, clarifying the importance of inulin intake in the morning on the beta-diversity and profile of the microbiota. However, in these experimental conditions, we did not control feeding and/or digestive speed; therefore, volume- and speed-controlled feeding systems may be required to determine the effect of feeding time.

The first meal after a long fast, most often breakfast, resets the phase of peripheral clock [52,53]. We recently demonstrated that cellobiose, a water-soluble dietary fiber, produces SCFAs, allowing them to reset the peripheral clock [21]. Taken together, these results suggest that the intake of inulin-containing foods in the morning may help reset the peripheral clock through SCFAs production.

Previous studies have reported that inulin consumption increases *Bifidobacteria* and *Akkermancia muciniphila* and decreases gram-positive cocci in humans and mice [47,48,49,50]. In this study, the gram-positive cocci *Streptococcus* and *Staphylococcus* decreased, but the *A. muciniphila* was not significantly changed. We considered that the degree of polymerization of inulin is one of the reasons that the results of this study differ from previous studies. In the structure of inulin, fructose is a monomer linked by 2–60 molecules with β-glycosidic bonds. The inulin used in this study had 16 fructose bonds (a degree of polymerization of 16) [34]. It has been reported that the influence on the microbiota is different depending on the degree of polymerization of inulin [48]. Therefore, the results may have been different with other degrees of polymerization.

*Streptococcus* is known to produce lactic acid [57,58], and *Streptococcus mutans* increases in the intestines of type 2 diabetes patients and is induced by a high-calorie diet [59]. Furthermore, *Staphylococcus aureus* is increased in obese patients, and *Staphylococcus* has a positive correlation with energy intake [60,61]. The SCFAs produced by inulin feeding increase the concentration of GLP-1 in the blood and promote insulin secretion [62]. Furthermore, SCFAs regulate insulin activity in adipose tissue through the GPR43 receptor [11]. Therefore, it has been suggested that inulin may be an effective food against diabetes. In fact, in rats and humans, inulin consumption inhibits blood glucose levels and lowers blood triglyceride levels and total cholesterol levels [35,63,64]. In this study, the species level was not fully detected, and blood glucose levels and triglycerides were not measured. If these factors were measured, we may have been able to clarify the relationship between the gut microbiota and glucose metabolism.

It has been reported that SCFAs produced by ingestion of water-soluble dietary fiber prevent fat accumulation in adipose tissue via GPR43 [11]. However, it has also been reported that water-soluble dietary fiber does not involve SCFAs and suppresses fatty acid accumulation itself. For example, water-soluble dietary fiber may form a highly viscous matrix in the small intestine, increase the viscosity of the small intestine, and then physically suppress fat absorption [65,66]. These reports should be considered when investigating the association between gut microbiota and lipid metabolism.

The analysis of the carbohydrate metabolism identified a significant association with the fructose and mannose metabolism in the morning inulin group under two meals per day but not under one meal per day. Inulin is a fructan polymerized with fructose. Therefore, it may be possible that fructose metabolism is more activated by inulin in the morning than in the evening. Furthermore, the production of SCFAs may be increased because fructose is metabolized in the morning. In addition, fructose metabolism may also be related to fasting time. PICRUSt is only a predictive tool. To determine accurate functional information of the related bacteria, metagenomic studies should be conducted. Additionally, the number of mice in each group should be increased to provide more accurate explanations regarding the microbiota and PICRUSt analysis.

## 5. Conclusions

In summary, inulin intake in the morning rather than in the evening affected the gut microbiota, promoted SCFAs production, and lowered the cecal pH. The difference between the morning and evening results was related to the fasting duration, suggesting that there may be a relationship between fasting duration and meal stimulation regarding control of the microbiota.

## Figures and Tables

**Figure 1 nutrients-11-02802-f001:**
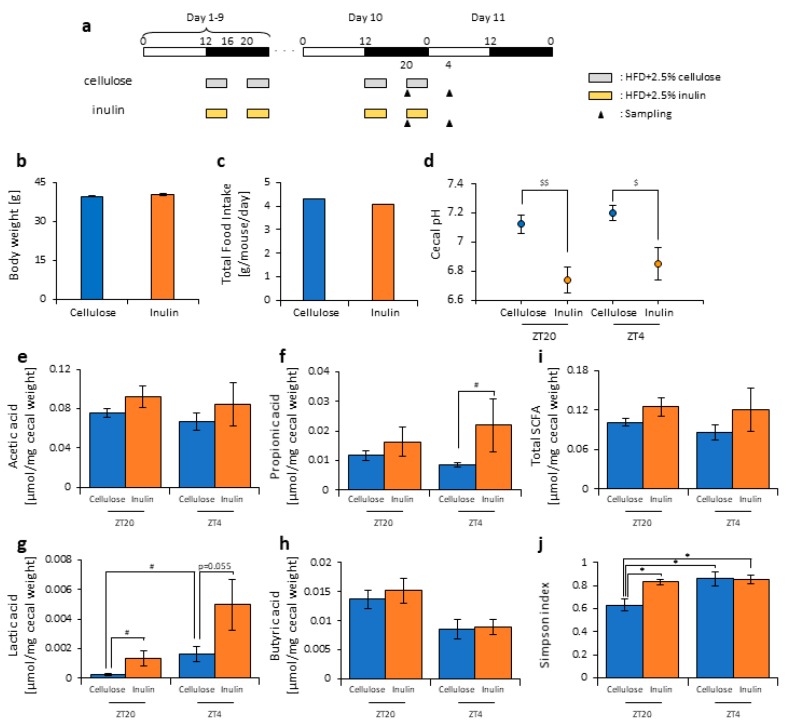
Inulin feeding decreased cecal pH and increased short-chain-fatty-acids. (**a**) Experimental schedule, where the white and black bars indicate environmental 12 h light and dark conditions, respectively. The gray bar indicates feeding with a high-fat-diet (HFD) and 2.5% cellulose. The yellow bar indicates feeding with HFD and 2.5% inulin. The black arrowhead indicates the sampling time. (**b**) Body weight before sampling. (**c**) Average daily food intake. (**d**) Cecal pH of mice housed for 10 days for each group. (**e**–**i**) The short-chain fatty acids (SCFAs) of mice, including (**e**) acetic acid, (**f**) propionic acid, (**g**) lactic acid, (**h**) butyric acid, and (**i**) total SCFAs. (**j**) Bacterial alpha diversity. Comparison of the Simpson index estimation of the 16S rDNA gene libraries at 97% similarity from the sequencing analysis. All values except (**c**) are represented as mean ± SEM (cellulose at ZT20 (n = 5) and 4 (n = 5); inulin at ZT20 (n = 5) and 4 (n = 5)). * *p* < 0.05, evaluated using the two-way ANOVA with Tukey’s post hoc test. $$ *p* < 0.01, $ *p* < 0.05, evaluated using the two-way ANOVA with Sidak’s post hoc test. # *p* < 0.05, evaluated using the Mann–Whitney test with a two-stage linear step-up procedure of the Benjamini, Krieger, and Yekutieli test for multiple comparisons.

**Figure 2 nutrients-11-02802-f002:**
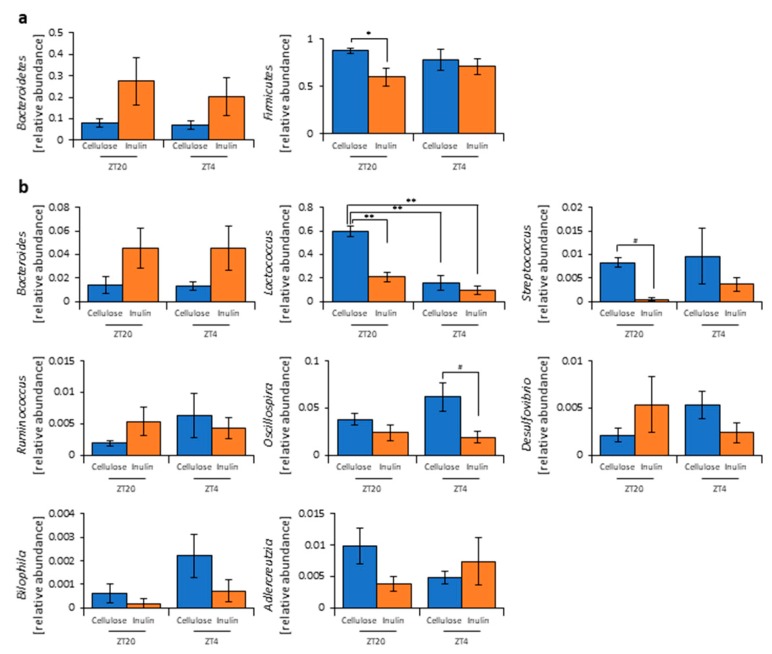
Inulin feeding changed the relative abundance of some bacteria. (**a**) Phylum level. (**b**) Genus level. All values are represented as mean ± SEM (cellulose at ZT20 (n = 5) and 4 (n = 5); inulin at ZT20 (n = 5) and 4 (n = 5)). ** *p* < 0.01, * *p* < 0.05, evaluated using the two-way ANOVA with Tukey’s post hoc test. # *p* < 0.05, evaluated using the Mann–Whitney test with a two-stage linear step-up procedure of the Benjamini, Krieger, and Yekutieli test for multiple comparisons.

**Figure 3 nutrients-11-02802-f003:**
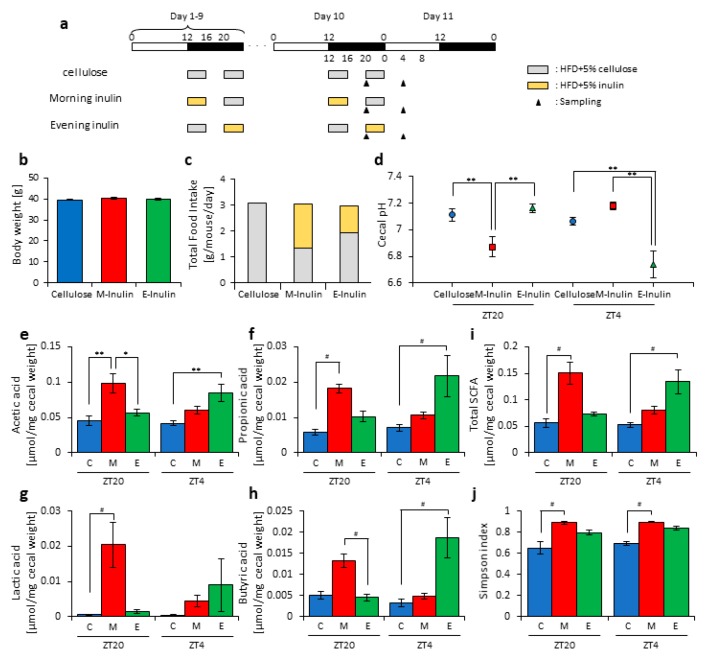
Morning inulin feeding decreased cecal pH and increased short-chain-fatty-acids more than evening inulin feeding. (**a**) Experimental schedule, where white and black bars indicate environmental 12 h light and dark conditions, respectively. The gray bar indicates feeding with a high-fat-diet (HFD) and 5% cellulose. The yellow bar indicates feeding with HFD and 5% inulin. The black arrowhead indicates the sampling time. (**b**) Body weight before sampling. (**c**) Average daily food intake. The gray bar indicates the average daily food intake of cellulose, and the yellow bar indicates the average daily food intake of inulin. (**d**) Cecal pH of mice housed for 10 days for each group. (**e**–**i**) SCFAs of mice, including (**e**) acetic acid, (**f**) propionic acid, (**g**) lactic acid, (**h**) butyric acid, and (**i**) total SCFAs. (**j**) Bacterial alpha diversity. Comparison of the Simpson index estimation of the 16S rDNA gene libraries at 97% similarity from the sequencing analysis. All values except (**c**) are represented as mean ± SEM (cellulose at ZT20 (n = 5) and 4 (n = 5); morning inulin at ZT20 (n = 5) and 4 (n = 5); evening inulin at ZT20 (n = 5) and 4 (n = 5)). ** *p* < 0.01, * *p* < 0.05, evaluated using the two-way ANOVA with Tukey’s post hoc test. # *p* < 0.05, evaluated using the Kruskal–Wallis test with Dunn post hoc test with a two-stage linear step-up procedure of the Benjamini, Krieger, and Yekutieli test for multiple comparisons. Cellulose, morning inulin, and evening inulin are C, M, or E, respectively.

**Figure 4 nutrients-11-02802-f004:**
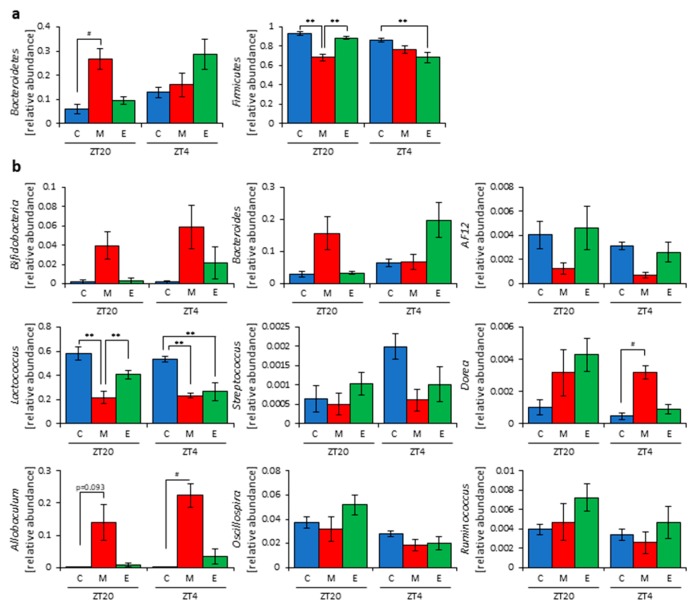
Morning inulin feeding changed the relative abundance of some bacteria. (**a**) Phylum level. (**b**) Genus level. All values are represented as mean ± SEM (cellulose at ZT20 (n = 5) and 4 (n = 5); morning inulin at ZT20 (n = 5) and 4 (n = 5); evening inulin at ZT20 (n = 5) and 4 (n = 5)). ** *p* < 0.01, evaluated using the two-way ANOVA with Tukey’s post hoc test. # *p* < 0.05, evaluated using the Kruskal–Wallis test with Dunn post hoc test with a two-stage linear step-up procedure of the Benjamini, Krieger, and Yekutieli test for multiple comparisons. Cellulose, morning inulin, and evening inulin are C, M, or E, respectively.

**Figure 5 nutrients-11-02802-f005:**
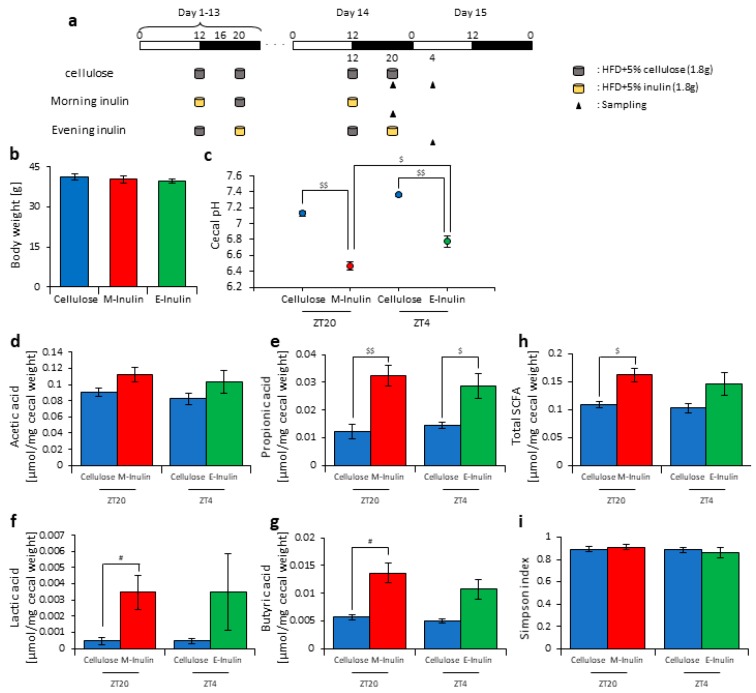
Morning inulin feeding decreased cecal pH and increased short-chain-fatty-acids more than evening inulin feeding under equivalent feeding conditions. (**a**) Experimental schedule, where white and black bars indicate environmental 12 h light and dark conditions, respectively. The gray cylinder indicates the 1.8 g of high-fat-diet (HFD) with 5% cellulose. The yellow cylinder indicates the 1.8 g of HFD with 5% inulin. The black arrowhead indicates the sampling time. (**b**) Body weight before sampling. (**c**) Cecal pH of mice housed for 14 days for each group. (**d**–**h**) SCFAs of mice, including (**d**) acetic acid, (**e**) propionic acid, (**f**) lactic acid, (**g**) butyric acid, and (**h**) total SCFAs. (**i**) Bacterial alpha diversity. Comparison of the Simpson index estimation of the 16S rDNA gene libraries at 97% similarity from the sequencing analysis. All values are represented as mean ± SEM (cellulose at ZT20 (n = 4) and 4 (n = 4); morning inulin (n = 5); evening inulin (n = 5)). $$ *p* < 0.01, $ *p* < 0.05, evaluated using the two-way ANOVA with Sidak’s post hoc test. # *p* < 0.05, evaluated using the Mann–Whitney test with a two-stage linear step-up procedure of the Benjamini, Krieger, and Yekutieli test for multiple comparisons.

**Figure 6 nutrients-11-02802-f006:**
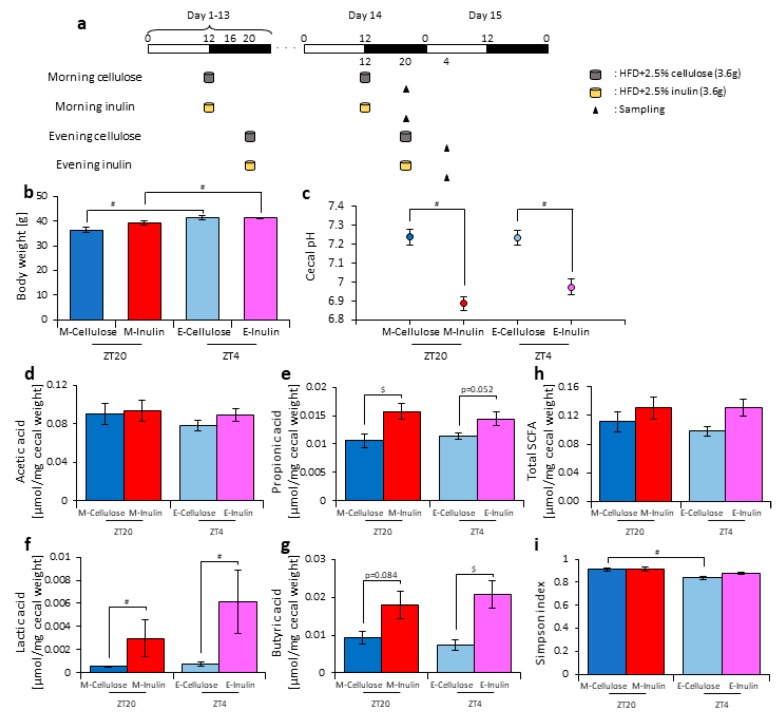
When fasting times are equal, the difference between morning and evening inulin feeding disappears. (**a**) Experimental schedule, where white and black bars indicate environmental 12 h light and dark conditions, respectively. The gray cylinder indicates the 3.6 g high-fat-diet (HFD) with 2.5% cellulose. The yellow cylinder indicates the 3.6 g of HFD with 2.5% inulin. The black arrowhead indicates the sampling time. (**b**) Body weight before sampling. (**c**) Cecal pH of mice housed for 14 days for each group. (**d**–**h**) SCFAs of mice, including (**d**) acetic acid, (**e**) propionic acid, (**f**) lactic acid, (**g**) butyric acid, and (**h**) total SCFAs. (**i**) Bacterial alpha diversity. Comparison of the Simpson index estimation of the 16S rDNA gene libraries at 97% similarity from the sequencing analysis. All values are represented as mean ± SEM (morning cellulose (n = 6); morning inulin (n = 6); evening cellulose (n = 6); evening inulin (n = 6)). $ *p* < 0.05, evaluated using the two-way ANOVA with Sidak’s post hoc test. # *p* < 0.05, evaluated using the Mann–Whitney test with a two-stage linear step-up procedure of the Benjamini, Krieger, and Yekutieli test for multiple comparisons. The table in (**j**) indicates the results using permutational multivariate analysis of variance (PERMANOVA). Morning cellulose, morning inulin, evening cellulose, or evening inulin are represented as M-cellulose, M-inulin, E-cellulose or E-inulin, respectively.

**Table 1 nutrients-11-02802-t001:** The relative abundance of some bacteria under the condition of two meals per day. (**a**). Phylum level. (**b**). Genus level.

**a. Phylum Level**						
**Bacterial**	**ZT20**			**ZT4**		
	**Cellulose**	**M-Inulin**	***p*-Value**	**Cellulose**	**E-Inulin**	***p*-Value**
*Actinobacteria*	0.0072 ± 0.0039	0.0103 ± 0.0081	0.7143	0.0690 ± 0.0647	0.0907 ± 0.0440	0.8254
*Bacteroidetes*	0.1535 ± 0.0463	0.2499 ± 0.0654	0.669	0.1647 ± 0.0467	0.3644 ± 0.1203	0.2085
*Deferribacteres*	0.0011 ± 0.0010	0.0005 ± 0.0004	0.9999	0.0002 ± 0.0002	0.0004 ± 0.0003	0.8413
*Firmicutes*	0.8221 ± 0.0467	0.6863 ± 0.0713	0.3232	0.7486 ± 0.0228	0.5294 ± 0.0865	0.0743
*Proteobacteria*	0.0111 ± 0.0033	0.0019 ± 0.0004	0.0159 ^#^	0.0152 ± 0.0030	0.0054 ± 0.0017	0.1905
*TM7*	0.0043 ± 0.0025	0.0004 ± 0.0003	0.0159 ^#^	0.0017 ± 0.0002	0.0001 ± 0.0001	0.0159 ^#^
*Verrucomicrobia*	0.0006 ± 0.0003	0.0473 ± 0.0219	0.1905	0.0003 ± 0.003	0.0132 ± 0.0117	0.1032
**b. Genus Level**						
**Bacterial**	**ZT20**			**ZT4**		
	**Cellulose**	**M-Inulin**	***p*-Value**	**Cellulose**	**E-Inulin**	***p*-Value**
*Bifidobacterium*	0.0022 ± 0.0018	0.0080 ± 0.0077	0.5556	0.0063 ± 0.0062	0.0088 ± 0.0044	0.6825
*Adlercreutzia*	0.0046 ± 0.0019	0.0022 ± 0.0004	0.2344	0.0047 ± 0.0014	0.0026 ± 0.0006	0.6428
*Bacteroides*	0.0463 ± 0.0175	0.1100 ± 0.0262	0.0993	0.0701 ± 0.0238	0.1953 ± 0.0627	0.1352
*Parabacteroides*	0.0010 ± 0.0003	0.0009 ± 0.0002	0.8247	0.0015 ± 0.0006	0.0010 ± 0.0003	0.6229
*AF12*	0.0075 ± 0.0023	0.0011 ± 0.0006	0.0435 ^#^	0.0052 ± 0.0013	0.0009 ± 0.0002	0.0317 ^#^
*Butyricimonas*	0.0004 ± 0.0001	0.0008 ± 0.0004	0.0317 ^#^	0.0005 ± 0.0003	0.0002 ± 0.0001	0.7937
*Odoribacter*	0.0017 ± 0.0003	0.0006 ± 0.0002	0.1795	0.0020 ± 0.0008	0.0003 ± 0.0001	0.0308 ^$^
*[Prevotella]*	0.0144 ± 0.0085	0.0303 ± 0.0098	0.2857	0.0118 ± 0.0064	0.0375 ± 0.0279	0.9762
*Staphylococcus*	0.0013 ± 0.0006	0.0002 ± 0.0001	0.0397 ^#^	0.0009 ± 0.0006	0.0005 ± 0.0002	0.8254
*Lactobacillus*	0.0294 ± 0.0230	0.0136 ± 0.0036	0.6825	0.1152 ± 0.0811	0.0631 ± 0.0323	0.873
*Lactococcus*	0.2862 ± 0.0453	0.0957 ± 0.0128	0.0159 ^#^	0.1691 ± 0.0446	0.0856 ± 0.0136	0.2857
*Streptococcus*	0.0034 ± 0.0016	0.0011 ± 0.0002	0.1545	0.0037 ± 0.0016	0.0019 ± 0.0010	0.357
*Clostridium*	0.0002 ± 0.0001	0.0001 ± 0.00005	0.3889	0.0014 ± 0.0012	0.0003 ± 0.0002	0.3889
*Dehalobacterium*	0.0018 ± 0.0002	0.0017 ± 0.0006	0.873	0.0010 ± 0.0001	0.0010 ± 0.0007	0.1746
*Coprococcus*	0.0049 ± 0.0009	0.0081 ± 0.0032	0.9999	0.0035 ± 0.0007	0.0019 ± 0.0005	0.1905
*Dorea*	0.0032 ± 0.0011	0.0032 ± 0.0012	0.9966	0.0041 ± 0.0031	0.0020 ± 0.0012	0.5089
*Roseburia*	0.0018 ± 0.0010	0.0053 ± 0.0038	0.9762	0.0005 ± 0.0004	0.0004 ± 0.0001	0.5635
*[Ruminococcus]*	0.0305 ± 0.0113	0.0384 ± 0.0105	0.6271	0.0327 ± 0.0129	0.0240 ± 0.0094	0.5957
*Oscillospira*	0.0803 ± 0.0131	0.0290 ± 0.0088	0.0148 ^$^	0.0600 ± 0.0173	0.0172 ± 0.0074	0.0404 ^$^
*Ruminococcus*	0.0096 ± 0.0019	0.0067 ± 0.0032	0.4961	0.0056 ± 0.0017	0.0025 ± 0.0010	0.161
*Allobaculum*	0.0017 ± 0.0007	0.1764 ± 0.0544	0.1905	0.0662 ± 0.0649	0.1669 ± 0.1094	0.5238
*Bilophila*	0.0013 ± 0.0003	0.0003 ± 0.0002	0.0159 ^#^	0.0011 ± 0.0005	0.0002 ± 0.0001	0.1111
*Desulfovibrio*	0.0037 ± 0.0011	0.0006 ± 0.0004	0.047 ^#^	0.0021 ± 0.0007	0.0019 ± 0.0002	0.371
*Akkermansia*	0.0006 ± 0.0002	0.0473 ± 0.0219	0.1905	0.0003 ± 0.0003	0.0132 ± 0.0117	0.1032

(**a**) Number of bacteria significantly changed by M-inulin/all number of bacteria well-detected = 2/7. Number of bacteria significantly changed by E-inulin/all number of bacteria well-detected = 1/7. ^#^
*p* < 0.05, evaluated using the Mann–Whitney test with a two-stage linear step-up procedure of the Benjamini, Krieger, and Yekutieli test for multiple comparisons. (**b**) Number of bacteria significantly changed by M-inulin/all number of bacteria well-detected = 7/24. Number of bacteria significantly changed by E-inulin/all number of bacteria well-detected = 3/24. ^$^
*p* < 0.05, evaluated using the two-way ANOVA with Sidak post hoc test. ^#^
*p* < 0.05, evaluated using the Mann–Whitney test with a two-stage linear step-up procedure of the Benjamini, Krieger, and Yekutieli test for multiple comparisons.

**Table 2 nutrients-11-02802-t002:** The relative abundance of some bacteria under the condition of one meal per day. (**a**). Phylum level. (**b**). Genus level.

**a. Phylum Level**						
**Bacterial**	**ZT20**			**ZT4**		
	**Cellulose**	**M-Inulin**	***p*-Value**	**Cellulose**	**E-Inulin**	***p*-Value**
*Actinobacteria*	0.0041 ± 0.0017	0.0364 ± 0.0199	0.0174 ^#^	0.0321 ± 0.0154	0.0871 ± 0.0380	0.1797
*Bacteroidetes*	0.0759 ± 0.0296	0.1372 ± 0.0337	0.3748	0.1024 ± 0.0275	0.1609 ± 0.0411	0.4069
*Deferribacteres*	0.0037 ± 0.0016	0.0018 ± 0.0006	0.5714	0.0007 ± 0.0004	0.0002 ± 0.00006	0.4459
*Firmicutes*	0.80427 ± 0.0240	0.7513 ± 0.0306	0.1797	0.7832 ± 0.0207	0.6906 ± 0.0373	0.1298
*Proteobacteria*	0.1119 ± 0.0161	0.0706 ± 0.0120	0.0799	0.0790 ± 0.0141	0.0568 ± 0.0102	0.4449
*Verrucomicrobia*	0.0001 ± 0.0001	0.0025 ± 0.0011	0.0606	0.0025 ± 0.0022	0.0042 ± 0.0036	0.5455
**b. Genus Level**						
**Bacterial**	**ZT20**			**ZT4**		
	**Cellulose**	**M-Inulin**	***p*-Value**	**Cellulose**	**E-Inulin**	***p*-Value**
*Bifidobacterium*	0.0004 ± 0.0003	0.0305 ± 0.0192	0.0043 ^##^	0.0266 ± 0.0151	0.0785 ± 0.0367	0.1775
*Adlercreutzia*	0.0036 ± 0.0017	0.0037 ± 0.0010	0.5714	0.0051 ± 0.0008	0.0041 ± 0.0008	0.3874
*Bacteroides*	0.0222 ± 0.0103	0.0327 ± 0.0078	0.4395	0.0149 ± 0.0051	0.0313 ± 0.0101	0.1801
*Parabacteroides*	0.0059 ± 0.0027	0.0032 ± 0.0010	0.8983	0.0048 ± 0.0012	0.0025 ± 0.0005	0.3874
*Butyricimonas*	0.0002 ± 0.00008	0.0001 ± 0.00004	0.6623	0.0003 ± 0.0001	0.0002 ± 0.0001	0.6591
*Odoribacter*	0.0015 ± 0.0003	0.0022 ± 0.0012	0.8983	0.0023 ± 0.0004	0.0031 ± 0.0007	0.3874
*[Prevotella]*	0.0066 ± 0.0031	0.0151 ± 0.0102	0.5541	0.0033 ± 0.0018	0.0056 ± 0.0045	0.9805
*Mucispirillum*	0.0037 ± 0.0016	0.0018 ± 0.0006	0.5714	0.0007 ± 0.0004	0.0001 ± 0.00006	0.4459
*Staphylococcus*	0.0005 ± 0.0002	0.00005 ± 0.00003	0.145	0.0009 ± 0.0004	0.00007 ± 0.00002	0.0022 ^##^
*Lactobacillus*	0.0080 ± 0.0031	0.0088 ± 0.0027	0.8983	0.1050 ± 0.0530	0.01225 ± 0.0046	0.1797
*Lactococcus*	0.2490 ± 0.0332	0.1468 ± 0.0205	0.094	0.3408 ± 0.0271	0.1824 ± 0.0494	0.0078 ^$$^
*Streptococcus*	0.0048 ± 0.0011	0.0013 ± 0.0002	0.0087 ^##^	0.0037 ± 0.0005	0.0023 ± 0.0006	0.0931
*SMB53*	0.0102 ± 0.0079	0.0130 ± 0.0078	0.9394	0.0467 ± 0.0224	0.0132 ± 0.0076	0.1688
*Dehalobacterium*	0.0017 ± 0.0004	0.0024 ± 0.0006	0.5628	0.0007 ± 0.0002	0.0021 ± 0.0008	0.132
*Blautia*	0.0004 ± 0.0001	0.0003 ± 0.00008	0.7381	0.0002 ± 0.0001	0.0003 ± 0.0001	0.1991
*Coprococcus*	0.0093 ± 0.0009	0.0116 ± 0.0026	0.7879	0.0028 ± 0.0005	0.0048 ± 0.0016	0.3874
*Dorea*	0.0017 ± 0.0004	0.0034 ± 0.0007	0.077	0.0011 ± 0.0003	0.0042 ± 0.0006	0.0016 ^$$^
*Roseburia*	0.00006 ± 0.00002	0.00005 ± 0.00003	0.9242	0.00007 ± 0.00002	0.0001 ± 0.00007	0.3398
*[Ruminococcus]*	0.0652 ± 0.0085	0.0509 ± 0.0082	0.3544	0.0299 ± 0.0059	0.0347 ± 0.0073	0.882
*Anaerotruncus*	0.0004 ± 0.0001	0.0001 ± 0.00005	0.1797	0.0001 ± 0.00004	0.0003 ± 0.0001	0.8312
*Oscillospira*	0.0473 ± 0.0076	0.0213 ± 0.0052	0.0043 ^$$^	0.0217 ± 0.0042	0.0158 ± 0.0022	0.6797
*Ruminococcus*	0.0082 ± 0.0009	0.0034 ± 0.0007	0.0012 ^$$^	0.0035 ± 0.0008	0.0020 ± 0.0003	0.7285
*Allobaculum*	0.0011 ± 0.0006	0.1122 ± 0.0562	0.0022 ^##^	0.0327 ± 0.0157	0.1703 ± 0.0513	0.0449 ^#^
*Catenibacterium*	0.0005 ± 0.0001	0.0003 ± 0.0001	0.3874	0.0004 ± 0.00009	0.0003 ± 0.0001	0.474
*Desulfovibrio*	0.0010 ± 0.0005	0.0011 ± 0.0007	0.9073	0.0006 ± 0.0002	0.0023 ± 0.0009	0.1001
*Citrobacter*	0.0030 ± 0.0011	0.0026 ± 0.0004	0.7879	0.0026 ± 0.0004	0.0015 ± 0.0003	0.0931
*Klebsiella*	0.0417 ± 0.0147	0.0328 ± 0.0138	0.5714	0.0412 ± 0.0105	0.0291 ± 0.0124	0.1797
*Akkermansia*	0.0001 ± 0.0001	0.0025 ± 0.0011	0.4545	0.0024 ± 0.0022	0.0041 ± 0.0036	0.5455

(**a**) Number of bacteria significantly changed by M-inulin/all number of bacteria well-detected = 1/6. Number of bacteria significantly changed by E-inulin/all number of bacteria well-detected = 0/6. ^#^
*p* < 0.05, evaluated using the Mann–Whitney test with a two-stage linear step-up procedure of the Benjamini, Krieger, and Yekutieli test for multiple comparisons. (**b**) Number of bacteria significantly changed by M-inulin/all number of bacteria well-detected = 5/28. Number of bacteria significantly changed by E-inulin/all number of bacteria well-detected = 4/28. ^$$^
*p* < 0.01, evaluated using the two-way ANOVA with Sidak post hoc test. ^##^
*p* < 0.01, ^#^
*p* < 0.05, evaluated using the Mann–Whitney test with a two-stage linear step-up procedure of the Benjamini, Krieger, and Yekutieli test for multiple comparisons.

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
