# Peer review of "Mice Microbiota Composition Changes by Inulin Feeding with a Long Fasting Period under a Two-Meals-Per-Day Schedule"

_nutrients, 2019, doi:10.3390/nu11112802_

Round 1

Reviewer 1 Report

The revision clarifies some points. Importantly the authors have maintained a conservative interpretation of the data and recognize the multifactorial nature of their studies. Circadian patterns are complex and involve a number of neuroendocrine changes along with metabolic changes and these are acknowledged.   The one point I request alteration is the use on line 96 of the term ‘worsen’ for the effect of high fat diet on the microbiota. It alters the microbiota, this is certain from many studies. Worsen implies a detrimental change and this may or might not be the case.   It is often stated that it is a more inflammatory-inducing or metabolism-altering microbiota and if the authors agree with this then reasons for the term worsen should be included. Notably many studies even now study microbiota changes under varios conditions, but cannot prove there is a connection between observed effects and the microbiome.   In this short revision, it should be clarified why this high fat diet was used.

Reviewer 2 Report

Most of the comments and suggestions were taken on board and the manuscript ids much improved However, it is still difficult to read and the concept of dysbiois is under considerable debate – given inter and intra individual differences – this term should not be applied.  Change or alteration to microbial populations is a more accurate reflection. The manuscript needs re writing throughout in terms of its English:  The text needs to be refined with a native English speaker

See below for examples – but this is recurring throughout the manuscript:

P1 Line 18 There are publications that look at the influence of inulin on the microbiota see https://doi.org/10.1038/s41598-018-34970-y and https://gut.bmj.com/content/68/8/1430 and https://doi.org/10.1093/ajcn/nqz001

P1 Line 33 Dysbiosis is not a good term to use here- inulin is part of the diet and dysbiosis suggests its not a good thing – alteration or change is more appropriate

P2 Line 62 rephrase “Such food components with a strong effect 62 on the microbiota are called prebiotics”. To  These prebiotics or food components can have a strong effect on the microbiota.

P2 Line 61-69 This paragraph needs re writing to tighten it up

P2 Line 80 Replace eating time with time of food intake

P2 Line Replace effects of the intake timing of water-soluble dietary fibers on the gut microbiota. With the dual effect on the microbiota of food type, particularly dietary fibers with time of its intake

P3 Line 96-97 Animals were fed a HFD in order to pre-dispose towards a non beneficial gut microbiota with the aim of enhancing any inulin benefiial effects to the gut microbiota.

Author Response

This manuscript is a resubmission of an earlier submission. The following is a list of the peer review reports and author responses from that submission.

Round 1

Reviewer 1 Report

see attached comm

General:

The article investigates the impact of inulin feeding on the gut microbiota. The authors assign relevance to this study by addressing the issue of timing effects on fiber intake. They examine this in food where 2 meals/day rather than ad lib were given to mice following long and short fasting periods (am- post activity and pm- low activity respectively). The authors examined microbial composition and functional changes after 10-14 days– specifically dietary inulin and microbial production of SCFA.  They followed up with a 1 –meal /day and examined similar changes.  They conclude that long fasting/high activity increased microbial processing an effect that was normalized in the latter.  

The interventions are done but no baseline measurements are recorded for the central core of this paper – microbial changes- and the control group (HFD only) is absent and this is a real pity. The n numbers are very low and not properly indicated for groups and treatments. I wouls like to see the Microbial data as central to the paper – it isn’t currently.  The significance table and PCA plots would be better as supplementary – it isn’t clear what the significance indicates.  The discussion does not add substance as it could in light of the information given – there are sonme broad statements but relevance should be attached to this data. I have added further comments below.

Introduction:

The text needs to be refined with a native English speaker. The concept of dysbiois is under considerable debate – given inter and intra individual differences – this term should not be applied. Change or alteration to microbial populations is a more accurate reflection.

Line 37 &38 and through out: Italics bacterial names

Line 37-39 There are many species of bacterial that are associated with CRC see Jahani-Sherafat et al. for a review – the most consistant among studies are Fusobacteria and F. nucleatum.

Line 45-50 SCFA are also signalling molecules through GPR43 (FFARs) please add this to intro

Line 56 Remove timing

Line 57 – remove dysbiois in favour of alterations/changes or some other term

Line 60-62 Tighten up the language here

Line 62 et al., should be in italics here and elsewhere

Line 67 Reference 23 and 24 are miscited also a recent paper on inulin and microbes would be useful to add here – Hoving et al., 2018 Scientific reports

Line 68-78 Perhaps describe the human aspect of late night gorging and low activity, in the next section it may be good to see how this relates to mice – given that night time is their high activity time.

Methodology

The study design appears novel. The illumina sequencing seems fine

However, number of animals is lacking and can only be assumed from PCA plots. Please add these as n>5 is not sufficient in power for this type of metabolite and 16S analysis.

The rationale for neither HFD selection was not indicated nor for cellulose (insoluble) is not provided. Was there a control group on HFD alone? It would be good to know if there was.  This would be valuble for analysis – alternatively a baseline pre-feed measurement would have added value to this study.

How was food intake measured?

ZT-20-0 should read ZT20-4

SCFA analysis by GC- there is no mention of range or standards? How was this done?

From 16S analysis another layer of data that is relevant relating to function relevant to carbohydrate metabolismis the use of PICRUSt used to investigate the abundances of gene families based on the 16S data and, from this data, you can infer functional alterations in the microbiome. From here the relative abundances of known bacterial genes involved in carbohydrate metabolism can be examined. This would add strength to the manuscript.

Results:

Figure 1 : well laid out but for K – the legend should sit with the graphs and it is not clear what the significance table represents - -this should be clarified

Figure S1a & b and S2a &b sand other supplementary figures hould be part of the main body as they is the central to the manuscript-

Figure 3 is fine but again the Table of significance isn’t clear nor where the significance lies and with which microbes? similarly for figure 4 &5

Discussion;

Line 428 Strong synchronization of the clock – this isn’t shown – although gene expression analyses (Per, clock, Bmal) from tissues could show this

Explain polymerization

Consider the pH changes and the alterations regionally and metabolically in the animal – could the pH changes be responsible for microbial changes and what would be the effect on gut health

Soluble fibres might alter lipid sequestering – FA accumulation an the means in which they affect the gut is not discussed – why might this be beneficial or not

How does this translate to humans given the over-tuned cycle – high activity but long starvation

ents

Reviewer 2 Report

This is a very focused study and that is a strength.   The study investigates whether the duration of fasting times alters the ‘response’ measured by short chain fatty acid generation and intestinal microbiota changes in response to dietary inulin, a fructose rich oligosaccharide commonly used by humans.   The study is very well constructed in multiple phases, following logically from establishing the model to alterations in timed feeding.   Timed feeding is a needed, but uncommonly investigated regulator of the response in the population of intestinal microbiota. This group has previous publications in metabolic and circadian changes caused by inulin and the present study helps to establish the potential contribution of the microbiota.   The conclusions are conservative which is appreciated.   The figures are outstanding and present the data well.   The writing is very clear and the manuscript well organized.  
